# ARGO: ARctic greenhouse Gas Observation metadata version 1

Judith Vogt[1,*], Martijn M. T. A. Pallandt[1,*], Luana S. Basso[1,*], Abdullah Bolek[1,*], Kseniia Ivanova[1,*], Mark Schlutow[1,*], Gerardo Celis[2], McKenzie Kuhn[3], Marguerite Mauritz[4], Edward A. G. Schuur[5], Kyle Arndt[6], Anna-Maria Virkkala[6], Isabel Wargowsky[6], and Mathias Göckede[1,*]

[1]Max Planck Institute for Biogeochemistry, Jena, Germany
[2]Department of Anthropology and Environmental Dynamics Program, University of Arkansas, Fayetteville, Arkansas, USA
[3]Department of Geography, University of British Columbia, Vancouver, Canada
[4]University of Texas at El Paso, El Paso, USA
[5]Center for Ecosystem Science and Society, Northern Arizona University, Flagstaff, USA
[6]Woodwell Climate Research Center, Falmouth, USA
[*]These authors contributed equally to this work

**Correspondence:** Judith Vogt (jvogt@bgc-jena.mpg.de)

**Abstract.** Our understanding of how rapid Arctic warming and permafrost thaw affect global climate dynamics is restricted by limited spatio-temporal data coverage due to logistical challenges and the complex landscape of Arctic regions. It is therefore crucial to make best use of the available observations, including the integrated data analysis across disciplines and observational platforms. To alleviate the data compilation process for syntheses, cross-scale analyses, earth system models, and remote sensing applications, we introduce the ARctic greenhouse Gas Observation metadata version 1 (ARGO), a new metadataset comprised of greenhouse gas observations from various observational platforms across the Arctic and boreal biomes within the polar region of the northern hemisphere. ARGO provides a centralised repository for metadata on carbon dioxide ($CO_2$), methane ($CH_4$), and nitrous oxide ($N_2O$) measurements, and is linked with an interactive online tool (https://www.bgc-jena.mpg.de/argo/). This tool offers prompt metadata visualisation for the research community. Here, we present the structure and features of ARGO, underscoring its role as a valuable resource for advancing Arctic climate research and guiding synthesis efforts in the face of rapid environmental change in northern regions. The ARGO meta-dataset is openly available for download at Zenodo (https://doi.org/10.5281/zenodo.13870390) (Vogt et al., 2024).

## 1 Introduction

The Arctic region is experiencing rapid warming, with temperatures rising nearly four times faster than the global average (Rantanen et al., 2022). This accelerated warming has profound implications for the Earth's climate system, as the Arctic plays a critical role in regulating global climate dynamics. Of particular concern is the thawing of permafrost, perennially frozen soils, which are estimated to contain carbon stocks of at least 1,700 Pg (Miner et al., 2022; Schaefer et al., 2014; Schuur et al., 2022). This enormous carbon reservoir is at risk of being partially released into the atmosphere upon thaw, triggering an accelerating feedback loop that would further amplify global warming. Simultaneously, ongoing warming within the high northern latitudes holds the potential to trigger substantial changes to permafrost ecosystem characteristics, including climate-

induced vegetation changes that may lead to shrubification (Mekonnen et al., 2021), or changes in hydrology (Andresen et al., 2020; Heslop et al., 2020) that alter greenhouse gas flux patterns.

Quantifying the current Arctic carbon budget requires a comprehensive monitoring network across this region. Furthermore, to facilitate accurate projections of its future evolution based on understanding the mechanisms that control carbon cycle dynamics, more in-situ data need to be integrated into process-based models (Watts et al., 2021; Natali et al., 2019). With Arctic landscapes being highly heterogeneous across spatial scales (Watts et al., 2021; Euskirchen et al., 2017; Virkkala et al., 2021), a large number of observation sites would be needed to resolve the pronounced variability in greenhouse gas processes (Pallandt et al., 2022). However, the vast size of the Arctic region, in combination with logistical challenges of harsh climate conditions and scarce infrastructure, has permitted the establishment of only sparse observational networks. This leads to data gaps and limited spatial and temporal data coverage, for example in Siberia, parts of Canada and in mountainous regions (Pallandt et al., 2022). Therefore, an inventory of research sites can aid in the identification of those gaps and provide guidance where new sites should be established.

To support data-driven syntheses and modeling activities with high-quality flux data, and facilitate the training and evaluation of earth system modeling and remote sensing applications, regional eddy covariance networks (AmeriFlux, EuroFlux, AsiaFlux) and global databases were established (Fluxnet2015, Fluxnet-$CH_4$, SRDB-V5, COSORE) (Baldocchi et al., 2001; Aubinet et al., 1999; Mizoguchi et al., 2009; Pastorello et al., 2020; Delwiche et al., 2021; Bond-Lamberty et al., 2020). Beyond these initiatives, several synthesis efforts that include the high northern latitude domain provide ecosystem or method-specific greenhouse gas data. These include, for example, the Arctic-Boreal $CO_2$ flux database (ABCflux) that combines data from eddy covariance towers and flux chambers for terrestrial ecosystems (Virkkala et al., 2022), the Boreal–Arctic Wetland and Lake Methane Dataset (BAWLD-$CH_4$) which synthesised chamber-based fluxes (Kuhn et al., 2021), the global lake and reservoir $CO_2$ synthesis for eddy covariance towers (Golub et al., 2023), or the Global River Methane Database (GRiMeDB) involving multiple non-eddy covariance techniques (Stanley et al., 2023).

While existing databases strongly contributed to advancing the understanding of climate change impacts on Arctic ecosystems (Ramage et al., 2024), all of them come with certain limitations with respect to comprehensive coverage. Most importantly, the synthesis efforts and databases listed above are either limited to one observational platform such as eddy covariance towers (Pastorello et al., 2020; Golub et al., 2023) or flux chambers (Jian et al., 2021; Bond-Lamberty et al., 2020; Kuhn et al., 2021), or are confined to single gases, for example carbon dioxide (Virkkala et al., 2022) or methane (Kuhn et al., 2021). Moreover, important ancillary information including site activity status or data coverage across different seasons is often lacking or difficult to extract from existing repositories, further complicating the tasks of evaluating network coverage and pinpointing gaps therein (Pallandt et al., 2022). Finally, in some cases information in databases is outdated, leading to contradictory metadata between repositories, and some (newer) sites not being listed.

Aiming at comprehensive metadata coverage across greenhouse gas species and platforms at high northern latitudes, we present the structure and the characteristics of the ARctic greenhouse Gas Observation metadata version 1 (ARGO). ARGO is a new meta-dataset, a compilation of metadata for sites where carbon dioxide ($CO_2$), methane ($CH_4$) or nitrous oxide ($N_2O$) have been measured. The focus of this new meta-dataset is to understand past and current greenhouse gas monitoring locations

by making existing metadata visible and searchable to identify temporal and spatial measurement coverage. The primary goal is not to provide a data-access portal, however, in some cases observational data are directly available through maintained databases, while in others data may only be available through links to related publications or contact with the site operators. The five observational platforms to monitor greenhouse gas processes considered here include (1) eddy covariance towers and (2) flux chambers, both operated in terrestrial or aquatic ecosystems (Fig. 1) and aiming to resolve processes from local to landscape scales. Furthermore, (3) atmospheric towers and (4) airborne measurements provide atmospheric observations that represent processes at regional to pan-Arctic scales. Finally, the estimation of (5) dissolved gas concentrations monitors highly relevant information that is available mostly for aquatic sites across a distributed Arctic network. ARGO aggregates greenhouse gas metadata specific to Arctic and boreal conditions within the Arctic polar region across these different platforms. To facilitate easy data selection and prompt visualisation, the meta-dataset is presented in an interactive online tool (https://www.bgc-jena.mpg.de/argo/) to provide an openly accessible and comprehensive overview for the research community. Our centralised repository of greenhouse gas metadata will guide future research efforts, ensuring that resources are directed towards filling critical gaps in our understanding of greenhouse gas observations in Arctic and sub-Arctic regions.

## 2 Methods

### 2.1 Framework

ARGO comprises metadata of study sites with greenhouse gas measurements from various observational platforms (Fig. 1) at high northern latitudes. The metadata consolidates basic information about location of the sites and their characteristics, measurement period, contact information, and links to scientific publications, published datasets, and repositories (Table 1). Further definitions of compiled data specific to each observational platform are given separately, with the full list of parameters included in ARGO for each observational platform given in Tables A1 to A5. With the help of ARGO, users can easily find out what type of greenhouse gas observations have been conducted where, in which years and by whom, and can address various questions related to high northern research sites. Information on different spatio-temporal and methodological categories is represented in ARGO:

– **Study domain**: This meta-dataset comprises sites within the borders of the Arctic polar region (Meredith et al., 2019), which primarily encompasses the Arctic biome and those parts of the boreal biome that are characterised by cryosphere elements such as permafrost and persistent winter season snow cover. The delineation of the domain has been defined somewhat flexibly on purpose, allowing for the inclusion of more southerly sites to reduce data gaps for certain biomes and regions, as further outlined in Section 3.

– **Land cover**: We categorise the ecosystem types represented by a study site as barren, cropland, forest, grassland, lake, ocean, reservoir, river, shrubland, tundra, urban, and wetland based on the associated publications or input from site operators and researchers. The lake category includes ponds and puddles, while the river category includes streams, ditches and canals. Where multiple ecosystem types apply to a site, we list all of them.

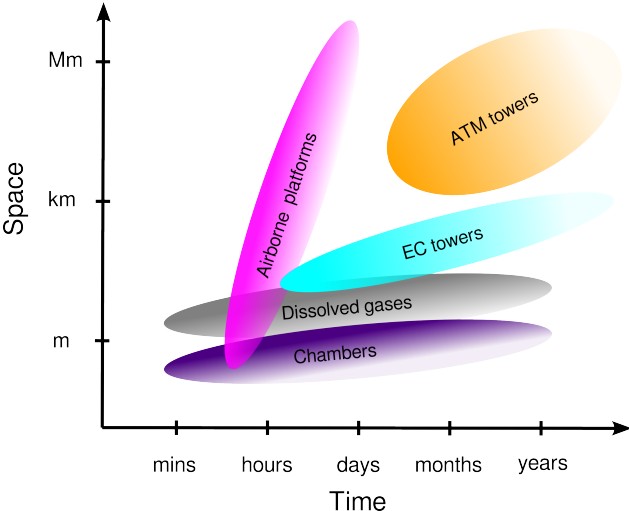

**Figure 1.** Stommel diagram showcasing temporal and spatial scales of observational platforms.

- **Timeframe**: Observations obtained at a specific location or area of interest studied between 1970 and 2024 were considered for this analysis.

- **Seasonality**: To categorise study periods, we distinguish between the growing season (labeled 'summer', months May–October) and the snow- and ice-covered season (labeled 'winter', months November–April). This definition does not necessarily align with the seasonal patterns of the different sites, but is used here for simplicity to differentiate between field visits taking place at different times of the year.

- **Gas species**: The greenhouse gases $CO_2$, $CH_4$, and $N_2O$ were considered.

- **Types of measurement**: Greenhouse gas measurements in this context include the assessment of atmospheric mole fractions, as well as vertical ecosystem-atmosphere, soil-atmosphere, or water-atmosphere fluxes, and concentration measurements of greenhouse gases dissolved in water.

## 2.2 Terrestrial versus aquatic systems

The study sites included in this database comprise a wide range of ecosystem types. We further divided them into two major categories, terrestrial and aquatic systems, because the processes governing greenhouse gas dynamics vary significantly between these systems, as do the potential controls on these variations. Terrestrial systems refer to all land-based observations, for example in forests, grasslands, and wetlands. Aquatic observations depict inland freshwater ecosystems, including lakes, rivers, reservoirs, ponds, streams, and ditches, but exclude marine sites. Given their large footprints that usually comprise mixed landscapes, observational data from atmospheric towers and airborne measurements were not assigned to specific

**Table 1.** Summary of the main site information in ARGO also shown in the online tool. More extensive tables outlining additional metadata can be found in the appendix (Tables A1 to A5). Measurement platforms are abbreviated with EC_Tower (eddy covariance towers), Chamber (chamber-based and ebullition measurements), Dissolved (dissolved gas measurements), ATM_Tower (atmospheric towers), and Airborne (drone- and airplane-based measurements).

| Column | Description |
| --- | --- |
| Site_Name | Name of the site |
| Site_ID | Abbreviation of site name, or network code (if applicable) |
| Latitude | Latitude position of the site (decimal degrees North; mean latitude of the outer bounds given for Airborne) |
| Longitude | Longitude position of the site (decimal degrees East; mean longitude of the outer bounds given for Airborne) |
| Ecosystem | Type of ecosystem of the site |
| Contact | Name of responsible person of the site |
| Contact_Email | E-mail address of responsible person |
| Years | Time period of measurements |
| Type | Type of observational platform (ATM_Tower, EC_Tower, Chamber, Dissolved, Airborne) |
| Country | Country of the site |
| Reference_Short | Short citation of publication (if applicable) |
| Additional_Information | Link to additional information (if available) |
| Data_Availability | Link of original database (if available) |

ecosystems, and for further analysis are assumed to represent terrestrial systems. The remaining platforms have terrestrial or aquatic contributions, or both.

## 2.3 Observational platforms

ARGO contains metadata from five distinct observational methods which cover the main techniques for in-situ greenhouse gas flux monitoring. Figure 1 displays the spatio-temporal scales associated with each of these platforms. The large range of scales, from minutes to years and from meters to thousands of kilometers, emphasises the need for a dataset that combines the strengths of different observational techniques for supporting comparative studies, data syntheses and modelling efforts.

### 2.3.1 Atmospheric towers

Sites belonging to this observational platform are equipped with atmospheric towers that collect data on atmospheric greenhouse gas mole fractions at regular intervals throughout the year. At most sites, towers are operated over a period of several years to decades. These measurements are taken within the continental boundary layer and integrate information from surface-atmosphere fluxes for large regions, with their footprints often covering areas of several 1000s of km$^2$, depending on the sampling height. As such, these data can be used to estimate fluxes when assimilated in atmospheric inverse modeling frameworks.

Measurements include both discrete flask air samples collected in the field and shipped to a laboratory for analysis, and continuous in-situ measurements using gas analysers installed at the sampling location. Typically, discrete air samples are collected in pairs of glass air flasks at weekly intervals, and are analysed for the main gases such as $CO_2$, $CH_4$, and $N_2O$, as well as minor trace gases and isotopic signals. Flask-based observations are mainly used to constrain long-term trends, and allow detailed attribution of the origin of air masses, for example, using isotopic analyses. Continuous in-situ measurements, typically available at hourly time steps, allow more detailed analysis of seasonal and short-term patterns, including variations in diurnal cycles.

Aside from the main site information summarised in Table 1, ARGO consolidates information on tower details, including ground elevation, tower height, and network provider. In addition, the database provides a summary of specific details about the time period of $CO_2$, $CH_4$ and $N_2O$ measurements. Information is provided on the sampling methodology, whether conducted in-situ or by flask sampling system, as well as details about the gas analysers and the sampling scheme. Further information on the availability of other measurements, such as carbon monoxide, carbon isotopes, and other greenhouse gases, is incorporated where available.

### 2.3.2 Eddy covariance towers

The eddy covariance method has been established to measure gas exchange between the biosphere and the atmosphere since the late 1980s (Baldocchi et al., 1988; Aubinet et al., 2012; Foken, 2017; Baldocchi, 2020). The technique is based on high-frequency instruments that continuously sample the turbulent fluctuations in wind speed and gas concentrations in the lower atmospheric boundary layer. After considering certain assumptions, net surface-atmosphere exchange fluxes for the sampled ecosystem can be derived based on the covariance between the vertical wind speed and gas concentration fluctuations.

In most cases, eddy covariance towers are deployed in a stationary setup and are accompanied by a range of ancillary measurements to resolve environmental parameters and local meteorology. Ecosystem fluxes are commonly aggregated to half-hourly averages corresponding to a specific dynamic footprint, with fetch sizes ranging between a few 100s of meters to a few kilometers depending on the tower height. Deployment times usually exceed one year, so that investigation of diurnal and seasonal cycles is possible, and for longer deployment times inter- and intra-annual variability can be monitored on an ecosystem scale. Due to maintenance and power supply limitations under harsh climate conditions, eddy covariance towers are rarely operated in the winter within the Arctic polar region (Pallandt et al., 2022).

The main metadata parameters for the eddy covariance towers are shown in Table 1, and further include information about measurement periods grouped by greenhouse gas ($CO_2$, $CH_4$ and $N_2O$), instrumentation types used for wind and greenhouse gas measurements, and complementary parameters that were measured. In addition, flux contributions from terrestrial and aquatic ecosystems are indicated.

### 2.3.3 Flux chambers

The chamber method involves the estimation of greenhouse gas fluxes within a sealed sample volume or headspace created with a chamber over soil or water. The concentration changes of gases within the headspace are monitored over time. Surface-

atmosphere fluxes are estimated based on measured concentration gradients and environmental conditions (temperature, pressure) commonly obtained from direct measurements. Generally, chamber measurements are used to capture instantaneous fluxes on small spatial scales (<1 m$^2$, Fig. 1). Approaches range from static chambers to automated systems, whereas static chambers are most common due to their low cost and simple deployment, even though they can only capture episodic snapshots of trace gas fluxes at selected sampling sites. Automated chambers are less common, require more resources and maintenance, but at the same time have the potential to deliver frequently repeated observations over months to decades. In ARGO, we did not differentiate between transparent and opaque chambers, which are typically used to distinguish between photosynthetic and respiratory $CO_2$ fluxes.

Apart from chamber measurements, this observational platform includes measurements taken to obtain ebullition fluxes. This pathway of gas release is especially relevant in freshwater environments. Ebullition measurements are typically conducted using bubble traps which consist of inverted funnels that are submerged and capture bursting bubbles rising from the sediment to the surface (Casper et al., 2000; Hamilton et al., 1994). Bubble traps are typically deployed over hours to days and fluxes are derived from volume and gas concentrations of the sampled bubbles. Ebullition fluxes represent a sporadic pathway of gas release, most prominent for $CH_4$, and are often assessed simultaneously with chamber measurements.

Site locations for chamber measurements are given as general areas of deployment for simplicity. This means that research areas are listed as sites, even though measurements may occur at many locations at the plot-scale within a specific research area. Metadata for this observational platform include more details about analysis techniques, chamber types used, and measurement periods. Chamber measurements were divided into terrestrial and aquatic measurements. In the latter case, chambers were commonly equipped with floats to avoid submersion.

### 2.3.4 Dissolved gases

This observational platform comprises measurements derived from analysed water samples, including water surface samples from aquatic sites as well as groundwater samples at terrestrial sites. Typically, dissolved gas concentrations are obtained through either in-situ or laboratory analyses with a greenhouse gas analyser. For some freshwater sites, water-air fluxes are derived from dissolved gas concentrations and the gas transfer velocity following Fick's law. A large number of techniques to derive gas transfer velocities exist and further discussion can be found in the literature (Klaus and Vachon, 2020; Wang et al., 2021).

Dissolved gas concentration measurements can be used to identify vertical surface-atmosphere exchange and lateral transport mechanisms, and particularly the input of carbon and nutrients to aquatic systems from surrounding landscapes or vice versa. Furthermore, fluxes derived from dissolved gas concentrations serve as an additional method in lieu of chamber measurements to determine sources and sinks of greenhouse gases in freshwater ecosystems.

In ARGO, additional information about techniques for sampling and analysis, instrumentation and measurement periods can be found for this observational platform.

### 2.3.5 Airborne platforms

Airborne observations provide a snapshot of greenhouse gas flux patterns or mole fractions over large areas. Airborne platforms (research manned aircrafts, or unmanned aerial vehicles (UAVs)) are commonly instrumented with flask samplers or gas analysers to sample greenhouse gases in the atmosphere, supported by a suite of meteorological instrumentation such as anemometers or temperature sensors. Manned aircrafts and larger UAVs have the capability to carry eddy-covariance instrumentation that can directly measure surface-atmosphere fluxes, while smaller UAVs, in particular, have limited payloads that mostly allow sampling of mole fractions of greenhouse gases in the lower atmosphere. In the latter case, surface-atmosphere fluxes can be constrained based on vertical and horizontal patterns in greenhouse gas mole fractions, or mass balance approaches when sampling, for example, the upwind and downwind sections of defined control volumes over the study area.

The resulting gas measurements from airborne platforms are typically campaign-based and not repeated regularly over extended periods of time. Airborne platforms are therefore highly suitable to complement stationary measurement platforms such as eddy covariance towers and chambers that provide high quality flux data though only for a fixed research area with limited spatial extent. Episodic airborne campaigns can overcome scaling challenges and allow for the assessment of the representativeness of stationary measurement devices in heterogeneous terrain.

With ARGO, we provide details about the measurement unit, number of flights conducted during campaigns, and ancillary measurements. For simplicity, we do not provide exact flight paths, but rather the outer bounds of areas covered during campaigns.

## 2.4 Data collection

The collection process covered a wide range of data sources. The version of the meta-dataset presented herein represents the status in June 2024. Metadata for eddy covariance sites were gathered predominantly from different flux databases (Fluxnet, AmeriFlux, AsiaFlux, ICOS, and NEON; Table 2) as described previously (Pallandt et al., 2022). In addition, metadata for atmospheric towers and airborne observations were gathered from various networks led by different institutions (NOAA, ICOS, JR-STATION, ECCC, GAW, ORNL DAAC, and HALO DB; Table 2). In addition, metadata embedded in existing syntheses (Virkkala et al., 2018; Virkkala and Miska, 2018; Virkkala et al., 2022; Kuhn et al., 2021; Stanley et al., 2023; Golub et al., 2023) were integrated or were extracted from scientific publications. The search for chamber and dissolved gas measurements was carried out in Google Scholar, Web of Science, ResearchGate, and eLibrary. Publications were searched using the following keywords: "carbon", "carbon dioxide", "methane", "CO2", "CH4", "greenhouse gas", "flux", "concentration", "dissolved", "Arctic", "permafrost", "tundra", "forest-tundra", "wetland", "lake", "pond", "river", "waterbody", and "reservoir". Furthermore, personal communication with site operators and researchers aided in the search for sites.

**Table 2.** Overview of general data sources for the different observational platforms of ARGO with descriptions and links to websites.

| Name | Observational Platforms | Description | Link |
|---|---|---|---|
| Fluxnet | Eddy covariance towers | Eddy covariance network (global) | https://fluxnet.org/ |
| AmeriFlux | Eddy covariance towers | Eddy covariance network (Americas) | https://ameriflux.lbl.gov/ |
| AsiaFlux | Eddy covariance towers | Eddy covariance network (Asia) | https://www.asiaflux.net/ |
| ICOS | Eddy covariance towers, Atmospheric towers | European Integrated Carbon Observation System | https://www.icos-cp.eu |
| NEON | Eddy covariance towers, Atmospheric towers | National Ecological Observation Network in United States of America | https://www.neonscience.org/data |
| ABCflux | Flux chambers | Synthesis of Arctic-boreal $CO_2$ fluxes | https://daac.ornl.gov/cgi-bin/dsviewer.pl?ds_id=1934 |
| BAWLD-$CH_4$ | Flux chambers | Synthesis of Arctic-boreal wetland and lake $CH_4$ fluxes | https://arcticdata.io/catalog/view/doi:10.18739/A2DN3ZX1R |
| GRiMeDB | Dissolved gases, Flux chambers | Synthesis of global river carbon fluxes and concentrations | https://portal.edirepository.org/nis/mapbrowse?packageid=knb-lter-ntl.420.2 |
| ECCC | Atmospheric towers | Atmospheric tower observation network by Environment and Climate Change Canada | https://www.canada.ca/en/environment-climate-change.html |
| GAW | Atmospheric towers | Global Atmosphere Watch Programme of World Meteorological Organization | https://community.wmo.int/en/activity-areas/gaw |
| NOAA/GML | Atmospheric towers | Global Atmospheric tower observation network by National Oceanic and Atmospheric Administration/Global Monitoring Laboratory | https://gml.noaa.gov |
| JR-STATION | Atmospheric towers | Japan–Russia Siberian Tall Tower Inland Observation Network by National Institute for Environmental Studies | https://www.cger.nies.go.jp/en/climate/pj1/tower/ |
| ORNL DAAC | Airborne platforms | Oak Ridge National Laboratory Distributed Active Archive Center for Biogeochemical Dynamics | https://daac.ornl.gov/get_data/ |
| HALO DB | Airborne platforms | Halo Database for Airborne Data | https://halo-db.pa.op.dlr.de/ |

## 3  Metadata overview

ARGO comprises metadata of sites with greenhouse gas measurements from five observational platforms (atmospheric and eddy covariance towers, chambers, dissolved gases, and airborne measurements) across the high northern latitudes gathered between 1970 to 2024 (Fig. 2). With the focus of the meta-dataset being placed on the Arctic polar region, about 83 % of sites are located above 60° N, and 44 % above the Arctic Circle. With 99 % of sites, the vast majority of the data stems from countries with land inside the Arctic Circle, including the USA (32 %), Russia (26 %), Canada (20 %), the Fennoscandian nations (16 %), and Greenland (4 %). The total latitudinal range of sites spans 42–83° N, with the southernmost sites being dominated by atmospheric towers with footprints extending over 1000s of km, and thus still covering large areas of the Arctic polar region. As mentioned above, the delineation of the ARGO domain was kept flexible to allow filling gaps with data from more southerly locations, leading to data contributions also from Estonia, Iceland, Ireland, Mongolia, Kazakhstan or Poland.

The temporal development of site coverage across monitoring networks is summarised in Fig. 3. After a slow start in network development following the establishment of the first monitoring sites in the 1970s to 1990s, the coverage of active greenhouse gas measurement sites within the Arctic polar region increased rapidly in the 2000s (Fig. 3). Until present, the network continues to grow, with the stagnation or even decrease in active site counts for the most recent years being an artifact associated with delays in processing the collected data and publishing it in databases and research articles. With regards to the activity of sites during different seasons of the year, two-thirds of the sites or studies investigated greenhouse gases exclusively during the growing season (eddy covariance: 73 %; chambers: 73 %; dissolved: 61 %; airborne: 100 %). At the same time, only about 30 % of the included sites are currently listed as being active year-round or during the winter months (eddy covariance: 27 %; chambers: 19 %; dissolved: 36 %; airborne: 0 %). All atmospheric towers are listed as operational year-round. The remaining minority of sites was investigated during various times of the year, or no information on the timing of site activity was available.

The distribution of sites across different ecosystems within ARGO is shown in Fig. 4. Lakes and rivers are well represented, especially for chamber-based and dissolved gas measurements. Tundra and wetlands are the most common terrestrial sites for chamber-based measurements. Forests and wetlands are most commonly targeted by the eddy covariance tower network.

The meta-dataset covers a comprehensive set of 62 atmospheric towers, almost 250 eddy covariance towers (regardless of operation time), and fourteen individual campaigns for airborne measurements. More than 1000 data points are listed for flux chamber measurements, close to 900 data points of dissolved greenhouse gases, with varying contributions from terrestrial and aquatic ecosystems (Fig. 5). ARGO is based on a research community of more than 400 scientists and provides their contact details (email address or links to datasets and websites, Table 1) where available. Three-quarters of the data are linked to 495 individual published studies, while the remaining data are either published in the form of datasets only or remain unpublished to date.

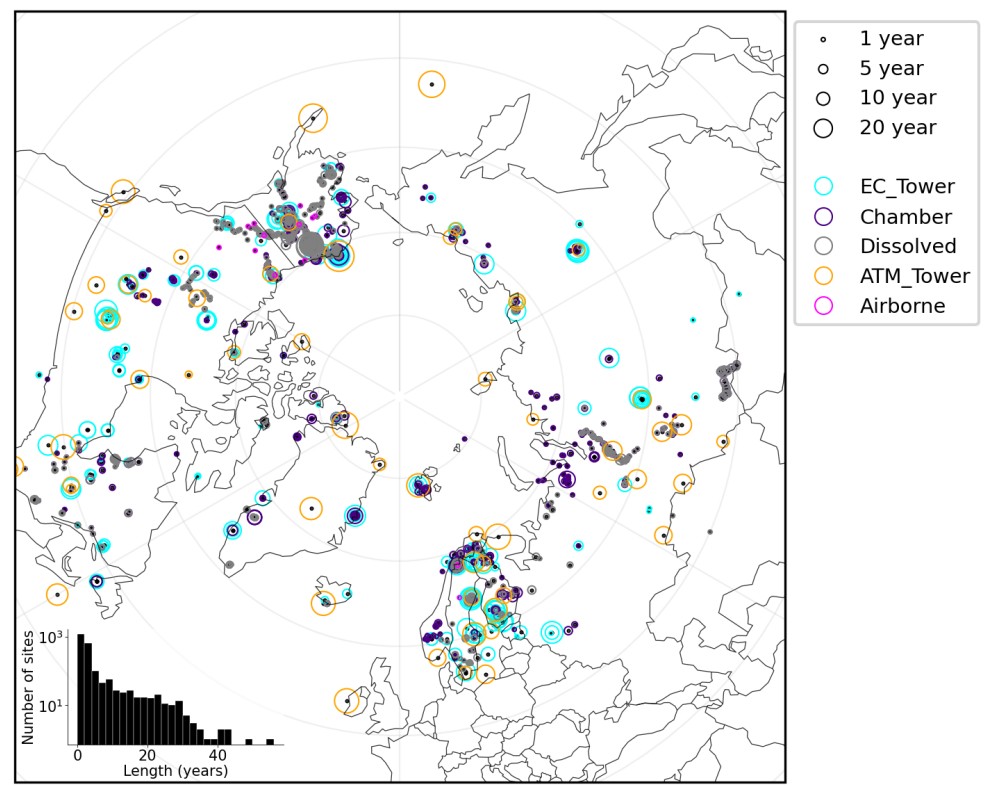

**Figure 2.** Distribution of study sites within the research domain. Colors of the circles represent sites for each observational platform included in the present database. For airborne measurements, the mean latitude and longitude of the outer bounds are shown. The size of the circles represents the number of years with measurements. The histogram at the bottom left shows the number of all sites binned in number of years with measurements. Measurement platforms are abbreviated as stated in Table 1.

## 4   Online mapping tool

The ARGO meta-dataset is visualised and made accessible online in the form of a map-based search tool (https://www.bgc-jena.mpg.de/argo/), offering an interactive map with site locations divided by observational platform. The conception of the online meta-database was initiated and strongly supported by two workshops held at the Arctic Data Center in Santa Barbara (California, USA), in 2018 (Parmentier et al., 2019), and with an ongoing data search the ARGO meta-database will be updated regularly in the future. With the online tool, users can explore the database, filter metadata by observational platform, select

measurement years, latitude and longitude zonal bands, terrestrial or aquatic ecosystem data, and greenhouse gases ($CO_2$, $CH_4$, $N_2O$). Additionally, users have the option to filter the metadata by country or seasonal activity. These filter options provide the user with a convenient tool to survey measurement sites and available datasets for various study purposes in the Arctic polar region. Selected variables are shown in a summary table (Table 1) providing key information for each site.

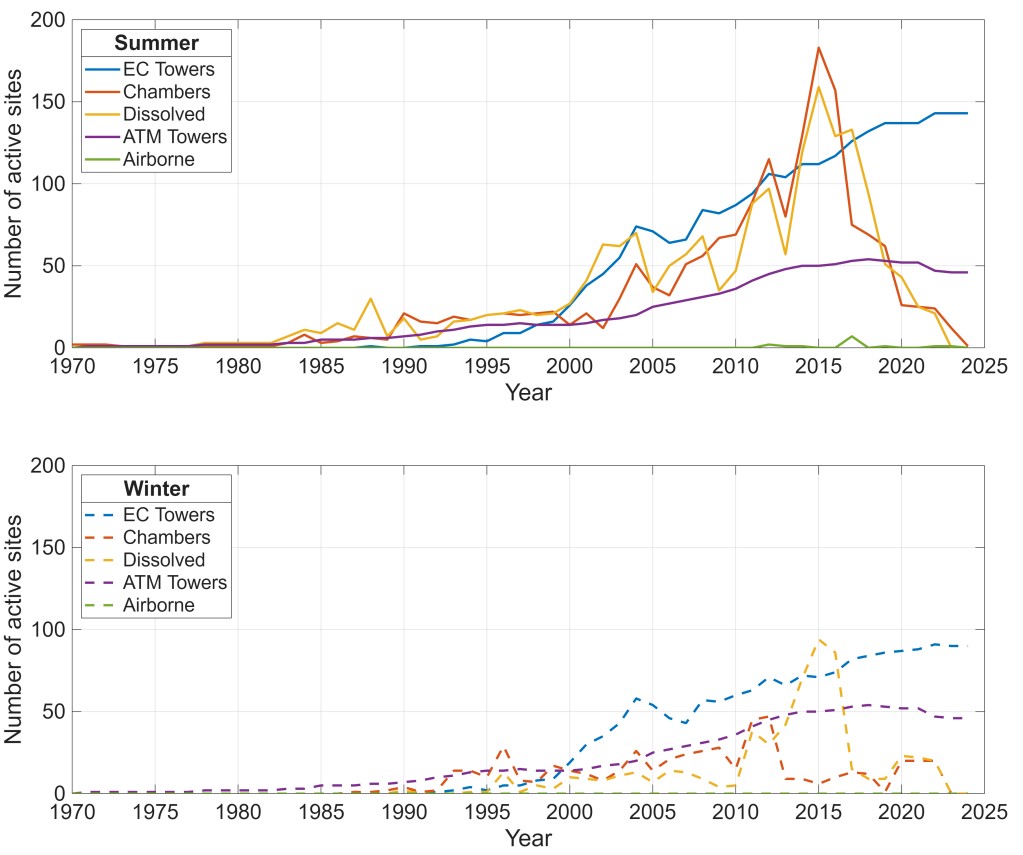

**Figure 3.** Cumulative temporal coverage of all available sites/studies shown separately for each observational platform. Measurement platforms are abbreviated as stated in Table 1. Please note that for EC and ATM towers, sites are considered to be operational until discontinuation is indicated by the operators, therefore data coverage remains high for these categories also in recent years. For all other categories, data availability relies on results or data being published, therefore time lags between measurement and being listed in this database lead to declining data coverage for the past decade. A distinction is made between summer (top) and winter activity (bottom). In cases where seasonal activity data are not available, only summer activity is assumed, except for atmospheric towers which are assumed to be active throughout the year.

For enhanced user accessibility, the metadata for all observational platforms along with a readme file are available for download in the form of compressed comma-separated files. Furthermore, users can download specific metadata tailored to their selected variables using the provided filters. Where relevant greenhouse gas flux or concentration data are publicly available, a link to the repository or dataset is provided within the meta-database (Data_Availability column, Table 1). In cases where data is published alongside a manuscript, for example as a supplementary file, this data could be accessed via the given reference (Reference_Short column, Table 1). Where additional information about a site is available, for example through other networks such as Fluxnet or Ameriflux, the link to this information is given as well (Additional_Information column, Table 1). In case

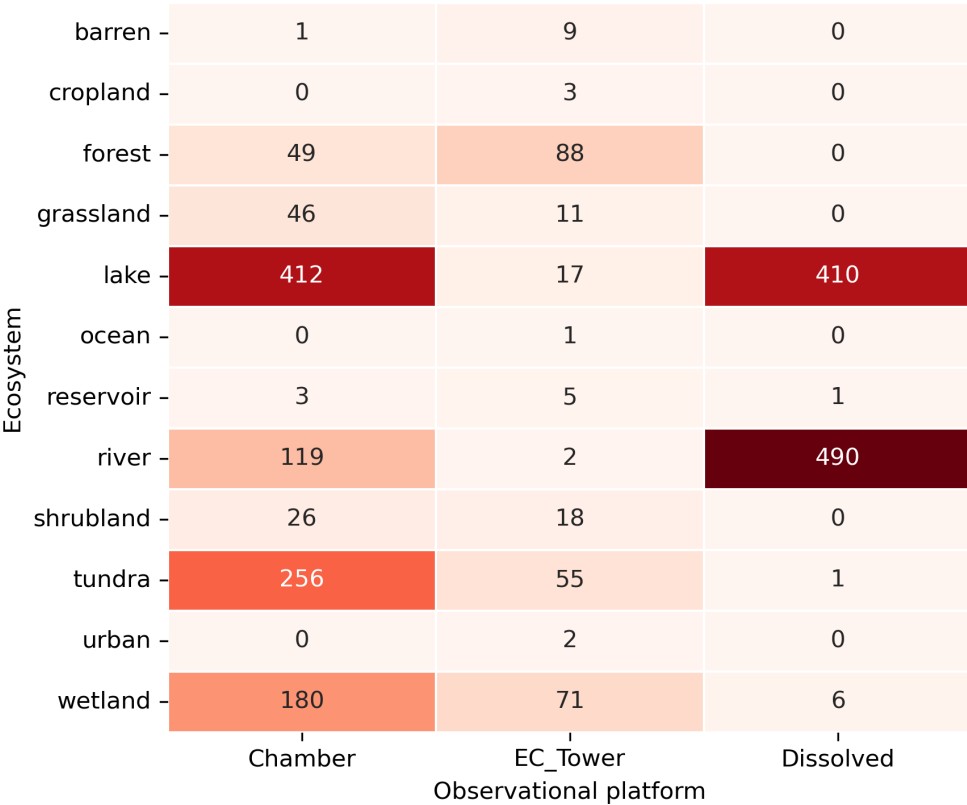

**Figure 4.** Counts of ecosystem types within the database. Note that ecosystem types are only given for eddy covariance towers, chamber-based, and dissolved gas measurements. Atmospheric tower footprints and airborne measurements are assumed to cover too large of an area for this analysis.

data remains unpublished, contact details are listed to initiate direct communication with members of the research community responsible for the site-specific data (Contact and Contact_Email columns, Table 1).

The online meta-database also offers a "How to Use" page, providing a detailed description of the web page functionalities and instructions on how to use the application. Furthermore, the "About" page provides comprehensive information regarding the scientific foundation of this project, including guidelines on citing the meta-database, references, and details about funding and the authors involved.

## 5 Data quality

To acquire comprehensive site-level metadata, and extend information provided by online databases, we conducted online surveys among principal investigators of Arctic flux sites, asking for information on, for example, exact times of measurements, instrumentation details, or ancillary measurements complementing the flux data (Pallandt et al., 2022). These surveys provide

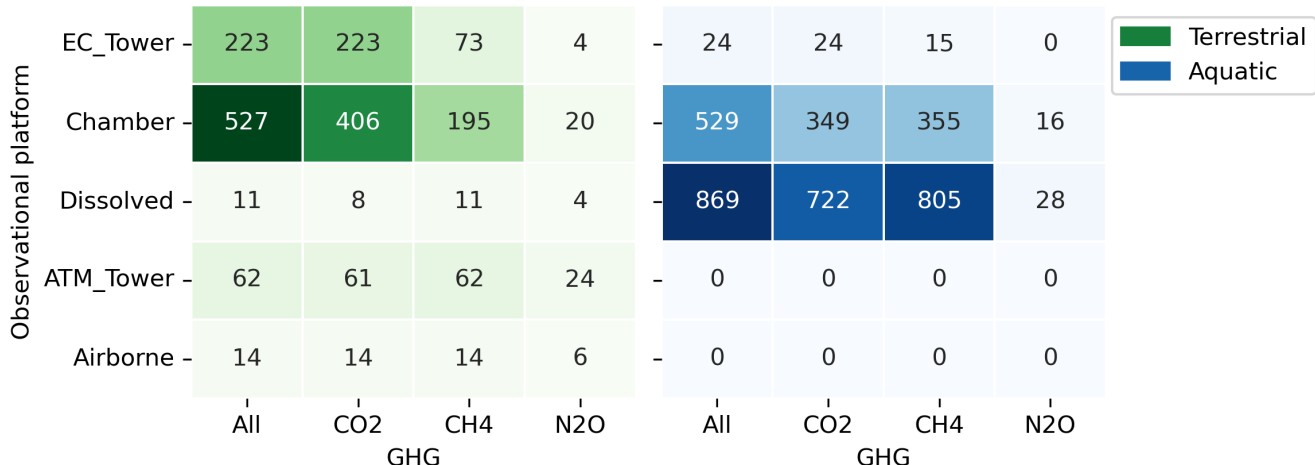

**Figure 5.** Number of sites for each observational platform and measured greenhouse gas (GHG) divided into terrestrial and aquatic sites. Measurement platforms are abbreviated as stated in Table 1.

a direct link between site operators or researchers and the metadata in ARGO. Both the detailed feedback given on different aspects of site operation as well as the option to discuss unclear information directly contributed to an improved accuracy of the relevant metadata in ARGO. As an example, database users have reported inconsistencies to database operators in the past, which improved the accuracy of the provided information. In addition, the metadata collection relied on existing peer-reviewed syntheses and published datasets. Since the metadata is visualised online, verification of the data collected by the network of researchers involved was possible in the past and continues to be easily possible.

To avoid displaying outdated information, the meta-database will be regularly checked and updated by the authors of this study in collaboration with site operators and researchers. The authors highly encourage site operators and researchers using the database to contact us with information, for example, about new sites, or updates regarding existing sites.

## 6  Data gaps and limitations

Site locations listed in this database are given as coordinates (latitude and longitude) with varying accuracy since sites were occasionally not geo-referenced especially in earlier studies, so that coordinates were approximated. In other cases, chamber and dissolved measurements repeated at close-by locations were consolidated into one single location to assure data usability and manageability within ARGO. This should be considered when high accuracy of site locations is required, for example, for merging in-situ observation with gridded remote sensing products.

From an ecosystem perspective, we identified low site representation across observational platforms especially for barren ecosystems, croplands, reservoirs, shrublands, and urban regions (Fig. 4). Lakes and rivers showed a decent representation across observational platforms, although the eddy covariance network would benefit from a larger number of towers targeted at

measuring water-atmosphere fluxes on an ecosystem scale. This becomes particularly important when considering that inland waterbodies cover a large area in Arctic-boreal regions (Pekel et al., 2016), and large emissions along with high uncertainties in current estimates of greenhouse gas budgets have been found (Ramage et al., 2024; Song et al., 2024).

To identify potential gaps in the spatial distribution of sites, the density of sites across observational platforms was visualised in Fig. 6 using ecoregions as defined by Olson et al. (2001). While the analysis identified well-represented ecoregions for $CO_2$, $CH_4$, and $N_2O$ measurements in western Scandinavia and northern and central Alaska, significant data gaps persist in other regions such as the eastern parts of Siberia, or central Canada. This is illustrated by the difference in average site density across large regions: Alaska showcases around 52 sites per 100,000 km$^2$, while Eastern Russia lags behind with barely 2 sites per 100,000 km$^2$. The distribution of sites separated by observational platform are shown in Fig. A1 to A3. Beyond these remote Arctic regions where access and logistics are particularly challenging, our database also confirms existing gaps in the network coverage in domains associated with generally low flux rates, such as barren tundra (Virkkala et al., 2018), or high-elevation areas within the Arctic-boreal domain (Pallandt et al., 2022).

Regarding temporal coverage, the growth of the network over the past decades as displayed in Fig. 3 has resulted in a current network of sites that facilitates pan-Arctic upscaling (Virkkala et al., 2025), and integrated trend analyses (See et al., 2024) across the Arctic-boreal domain. However, only very few sites were kept active continuously over two decades or more (Pallandt et al., 2024). Therefore, the analysis of long-term trends is restricted to a few pinpoints across the map, and information that goes beyond the turn of the century is particularly scarce. Moreover, wintertime coverage lags behind the summertime observations by about 20 years (Fig. 3). As a result, large coverage gaps outside the growing season particularly for non-$CO_2$ gases exist. For $CH_4$, terrestrial measurements are largely restricted to the growing season, and excluding the atmospheric towers, the database currently just lists 49 entries for year-round or wintertime datasets. This gap is partly balanced by a quite large number (292) of wintertime dissolved gas measurements. However, these are mostly coming from experiments focusing on small regions in Alaska and Russia. For $N_2O$, only 7 % of the data cover the cold season.

Regarding gas species, with only 93 data points $N_2O$ is the least covered greenhouse gas within ARGO, and substantial temporal and spatial gaps still need to be filled: Outside the atmospheric tower network (24 towers), more than 50 % of the available $N_2O$ data are provided by sites in Fennoscandia, leading to a strong regional focus, and large gaps in most other Arctic regions. With particularly Yedoma soils having been identified as relevant soil nitrogen pools (Strauss et al., 2022), observed (Marushchak et al., 2021) and potential future emissions of $N_2O$ (Strauss et al., 2024) could contribute a substantial fraction to the net greenhouse gas budget of the Arctic-boreal domain. $N_2O$ therefore needs to be monitored more closely.

Concerning observational platforms, airborne observations provide a very valuable addition to the largely stationary network. Even though only few datasets are available for this platform, these consist of extended flight legs each, covering large areas with very detailed and information-rich observations. From the pan-Arctic perspective, the main gap consists in the uneven spatial distribution: From the 14 airborne datasets currently listed in ARGO, just one is not focusing on Alaska and northwestern Canada. Moreover, no campaigns were conducted during wintertime. Since spatially-extensive flight legs provide information on flux variability and site conditions from landscape- to regional-scales, they are invaluable for interpreting the

spatial representativeness of data sources with smaller footprints. Accordingly, an extended coverage would boost our ability for insights on pan-Arctic carbon cycle processes through assessments integrating across platforms.

Combining measurements of several observational platforms provides an advantage for better understanding of greenhouse gas dynamics across both spatial and temporal scales. Including eddy covariance, flux chambers, and dissolved gas measurements, we searched for sites co-located within a 300 m radius, and found 368 clusters of sites where at least two of these observational platforms were used within close proximity. In 91 % of these clusters, flux chamber and dissolved gas measurements were conducted with a majority of those in aquatic systems. Only at 2 % of site clusters all three observational platforms

were used, but examples where these measurements occurred simultaneously are scarce (Erkkilä et al., 2018; Jansen et al., 2020; Jammet et al., 2017). Therefore, studies combining simultaneous measurements with different observational platforms could overcome scaling issues, and improve our understanding of greenhouse gas dynamics across small scales.

       The current trend of relocating research activities operated by Western scientists from Russian research sites to other regions as a result of ongoing conflicts between Russia and Ukraine further degrades the number of active sites in Russia, with signif-

icant impacts on Arctic science overall (López-Blanco et al., 2024; Schuur et al., 2024). For the pan-Arctic eddy-covariance network, Schuur et al. (2024) quantified a loss of spatial representativeness from 0.55 to 0.36 (minus 35 %) linked to the missing access to 27 stations situated in Russia. Even a targeted investment into new sites, for example in North America, could only make up for about one third of this information loss. This emphasises the need to, when the time is right, develop strategies to keep Russian sites operational, and facilitate data exchange and communication for the benefit of Arctic climate research.

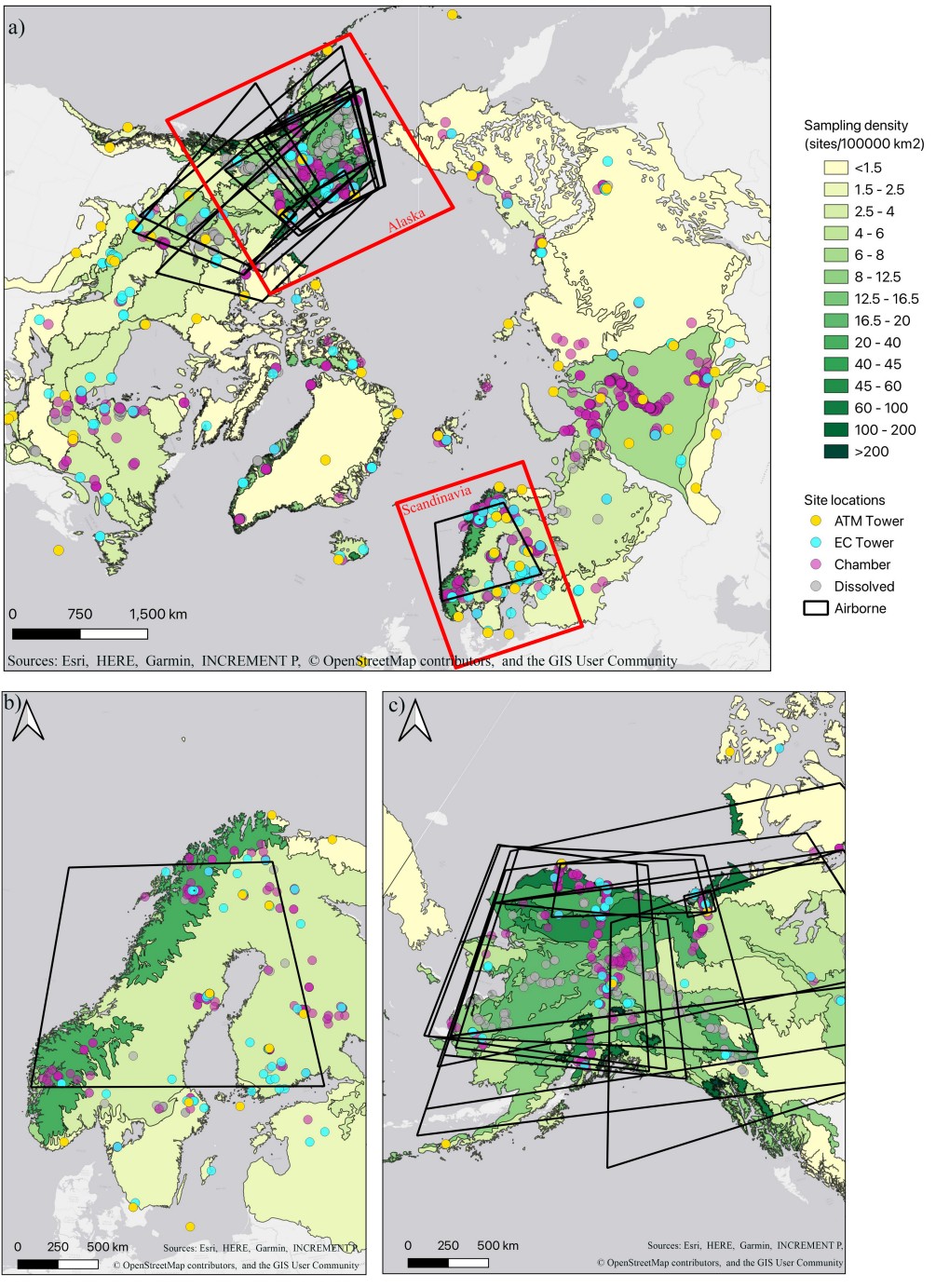

**Figure 6.** Density of study sites within ecoregions across high northern latitudes (a), with zoomed view for Scandinavia (b), and Alaska (c). Numbers are normalised to ecoregion area. Ecoregions were defined based on Olson et al. (2001). Measurement platforms are abbreviated as stated in Table 1.

## 7 Data availability

The ARGO metadata version 1 presented in this study has been published to the Zenodo repository (https://doi.org/10.5281/zenodo.13870390) under license CC-BY-4.0 (Vogt et al., 2024). The interactive tool visualising the latest version of the metadata can be accessed online (https://www.bgc-jena.mpg.de/argo/), and metadata download is also facilitated from that platform.

## 8 Code availability

The code to reproduce the online tool can be found in a public GitLab repository (https://git.bgc-jena.mpg.de/ipas/argo) and at https://doi.org/10.5281/zenodo.12795381.

## 9 Conclusions

The novel meta-database ARGO comprises information on temporal and spatial extent as well as technical and ancillary information on five different observational platforms that provide information on greenhouse gas processes within the Arctic-boreal domain. The metadata within ARGO can be used as a basis to support the planning and execution of studies aimed at synthesising functional relationships governing greenhouse gas exchange processes, and aggregating greenhouse gas budgets, at the pan-Arctic scale or for selected sub-regions. In addition, the ARGO meta-database provides an easy-to-use online tool to visualise data coverage and identify gaps therein, also facilitating the selection of user-defined subsets of data by applying filters. This online mapping tool can therefore guide future research activities towards strengthening observational capacities by filling crucial data gaps.

As data scarcity remains a major obstacle to data-driven assessments of carbon budgets in the Arctic-boreal study domain, improving visibility and access to distributed and heterogeneous data sources will reduce discrepancies in observation-based carbon budget estimates between synergy studies. In this context, ARGO aims at expediting the search for existing data, and maximising the available database, for ongoing and future synergy studies. With this service, ARGO supports the Arctic-boreal research community to better understand greenhouse gas cycle processes in the northern study domain, which is highly important for assessments of global greenhouse gas dynamics and future climate projections.

The ARGO meta-dataset described and shown here is a frozen-in-time version that is accessible as outlined in Section 7. Future maintenance of ARGO will be carried out by several international research groups, as reflected by the affiliations given for our author list, with this large community ensuring long-term and continuous support. Responsible team members will update the data tables on a fixed schedule several times per year, with new information also building on feedback that we will request from the Arctic research community through newsletters, prompting colleagues to enlist also metadata for yet unpublished studies. As a consequence, the 'active' version of the database as reflected in the online tool is expected to quickly deviate from the frozen-in-time version described herein, and the latest version of ARGO can be found in the online tool as continuous updates will integrate new sites and studies becoming available in the future.

**Table A1.** Full list of metadata descriptors for atmospheric towers.

| Column | Description |
| --- | --- |
| Type | Type of observational platform (here ATM_Tower for atmospheric towers) |
| Site_Name | Name of the site |
| Site_ID | Abbreviation of site name (as used in data repositories) |
| Latitude | Latitude of the site (in decimal degrees North) |
| Longitude | Longitude of the site (in decimal degrees East) |
| Country | Country of the site |
| Contact | Name of person responsible for site/data |
| Contact_Email | E-mail address of responsible person |
| Data_Availability | Link to data source or repository (if available) |
| Additional_Information | Link to additional information, e.g. description about the tower |
| Ground_Elevation | Elevation of sample collection above ground (m) |
| Tower_Height | Height of measurement tower above ground (m) |
| Network_Provider | Provider of the data |
| Gas_Analyser | Information on instrument used for gas analysis |
| Insitu_Parameters | List of gases sampled in-situ |
| Flask_Parameters | List of parameters analysed from flask sample |
| Sampling_Scheme | Methodological details about sampling frequency |
| GHG | Greenhouse gases measured: $CO_2$ (carbon dioxide), $CH_4$ (methane), $N_2O$ (nitrous oxide) |
| Start_CO2 | First year of $CO_2$ measurements |
| End_CO2 | Last year of $CO_2$ measurements |
| Start_CH4 | First year of $CH_4$ measurements |
| End_CH4 | Last year of $CH_4$ measurements |
| Start_N2O | First year of $N_2O$ measurements |
| End_N2O | Last year of $N_2O$ measurements |
| Season_Activity | Measurement period of the year |
| Terrestrial | Flag for land-based observations/fluxes |
| Aquatic | Flag for inland freshwater observations/fluxes |

## Appendix A: Metadata descriptors

A full list of parameters contained in the ARGO metadata is presented in Tables A1 to A5. Since each observational platform has slightly different parameters, the descriptors are given in separate tables. The spatial distribution of study sites for each observational platform are given in Fig. A1 to A3.

**Table A2.** Full list of metadata descriptors for eddy covariance towers.

| Column | Description |
|---|---|
| Type | Type of observational platform (here EC_Tower for eddy covariance towers) |
| Site_Name | Name of the site |
| Site_ID | Abbreviation of site name (as used in data repositories) |
| Latitude | Latitude of the site (in decimal degrees North) |
| Longitude | Longitude of the site (in decimal degrees East) |
| Country | Country of the site |
| Contact | Name of person responsible for site/data |
| Contact_Email | E-mail address of responsible person |
| Data_Availability | Link to data source or repository (if available) |
| Reference | Full citation of publication (if applicable) |
| Reference_Short | Short citation of publication (if applicable) |
| Additional_Information | Link to additional information, e.g. description about the tower |
| Anemometer | Anemometer model (if available) |
| Gas_Analyser | Information on instrument used for gas analysis |
| Power | Type of power source |
| Ecosystem | List of ecosystems applicable to the site: barren, cropland, forest, grassland, lake, ocean, reservoir, river, shrubland, tundra, urban, wetland |
| GHG | Greenhouse gases measured: $CO_2$ (carbon dioxide), $CH_4$ (methane), $N_2O$ (nitrous oxide) |
| Complementary_Measurements | List of other variables measured |
| Start_CO2 | First year of $CO_2$ measurements |
| End_CO2 | Last year of $CO_2$ measurements |
| Start_CH4 | First year of $CH_4$ measurements |
| End_CH4 | Last year of $CH_4$ measurements |
| Start_N2O | First year of $N_2O$ measurements |
| End_N2O | Last year of $N_2O$ measurements |
| Season_Activity | Measurement period of the year |
| Terrestrial | Flag for land-based observations/fluxes |
| Aquatic | Flag for inland freshwater observations/fluxes |

**Table A3.** Full list of metadata descriptors for flux chambers.

| Column | Description |
| --- | --- |
| Type | Type of observational platform (here Chamber for chamber-based and ebullition measurements) |
| Site_Name | Name of the site |
| Latitude | Latitude of the site (in decimal degrees North) |
| Longitude | Longitude of the site (in decimal degrees East) |
| Country | Country of the site |
| Contact | Name of person responsible for site/data |
| Contact_Email | E-mail address of responsible person |
| Data_Availability | Link to data source or repository (if available) |
| Reference | Full citation of publication (if applicable) |
| Reference_Short | Short citation of publication (if applicable) |
| Analysis_Technique | Sample analysis: In-situ or Lab (ex-situ laboratory-based analysis) |
| Gas_Analyser | Information on instrument used for gas analysis |
| Chamber_Type | Type of chamber: Manual chamber, Automatic chamber, or Ebullition trap |
| Ecosystem | List of ecosystems applicable to the site: barren, cropland, forest, grassland, lake, ocean, reservoir, river, shrubland, tundra, urban, wetland |
| Ecosystem_Details | Description of ecosystem of the site |
| GHG | Greenhouse gases measured: $CO_2$ (carbon dioxide), $CH_4$ (methane), $N_2O$ (nitrous oxide) |
| Start_Year | First year of measurements |
| End_Year | Last year of measurements |
| Season_Activity | Measurement period of the year |
| Terrestrial | Flag for land-based observations/fluxes |
| Aquatic | Flag for inland freshwater observations/fluxes |
| Comment | Notes and comments |

**Table A4.** Full list of metadata descriptors for dissolved gases.

| Column | Description |
| --- | --- |
| Type | Type of observational platform (here Dissolved for water-based gas concentration measurements) |
| Site_Name | Name of the site |
| Latitude | Latitude of the site (in decimal degrees North) |
| Longitude | Longitude of the site (in decimal degrees East) |
| Country | Country of the site |
| Contact | Name of person responsible for site/data |
| Contact_Email | E-mail address of responsible person |
| Data_Availability | Link to data source or repository (if available) |
| Reference | Full citation of publication (if applicable) |
| Reference_Short | Short citation of publication (if applicable) |
| Ecosystem | List of ecosystems applicable to the site: barren, cropland, forest, grassland, lake, ocean, reservoir, river, shrubland, tundra, urban, wetland |
| Ecosystem_Details | Description of ecosystem of the site |
| GHG | Greenhouse gases measured: $CO_2$ (carbon dioxide), $CH_4$ (methane), $N_2O$ (nitrous oxide) |
| Start_Year | First year of measurements |
| End_Year | Last year of measurements |
| Season_Activity | Measurement period of the year |
| Terrestrial | Flag for land-based observations/fluxes |
| Aquatic | Flag for inland freshwater observations/fluxes |
| Comment | Notes and comments |

**Table A5.** Full list of metadata descriptors for airborne platforms.

| Column | Description |
| --- | --- |
| Type | Type of observational platform (here Airborne) |
| Site_Name | Name of the site |
| Latitude | Northern latitude of the overflown area (decimal degrees North) |
| Latitude_S | Southern latitude of the overflown area (decimal degrees North) |
| Longitude | Western longitude of the overflown area (decimal degrees East) |
| Longitude_E | Eastern longitude of the overflown area (decimal degrees East) |
| Country | Country of the site |
| Contact | Name of person responsible for site/data |
| Contact_Email | E-mail address of responsible person |
| Data_Availability | Link to data source or repository (if available) |
| Reference | Full citation of publication (if applicable) |
| Reference_Short | Short citation of publication (if applicable) |
| Number_Of_Flights | Number of flights per campaign |
| Complementary_Measurements | List of other variables measured |
| Measurement_Unit | Type of measurement unit (aircraft or unmanned aerial vehicle) |
| GHG | Greenhouse gases measured: $CO_2$ (carbon dioxide), $CH_4$ (methane), $N_2O$ (nitrous oxide) |
| Campaign_Start | First day of measurements |
| Campaign_End | Last day of measurements |
| Season_Activity | Measurement period of the year |
| Terrestrial | Flag for land-based observations/fluxes |
| Aquatic | Flag for inland freshwater observations/fluxes |

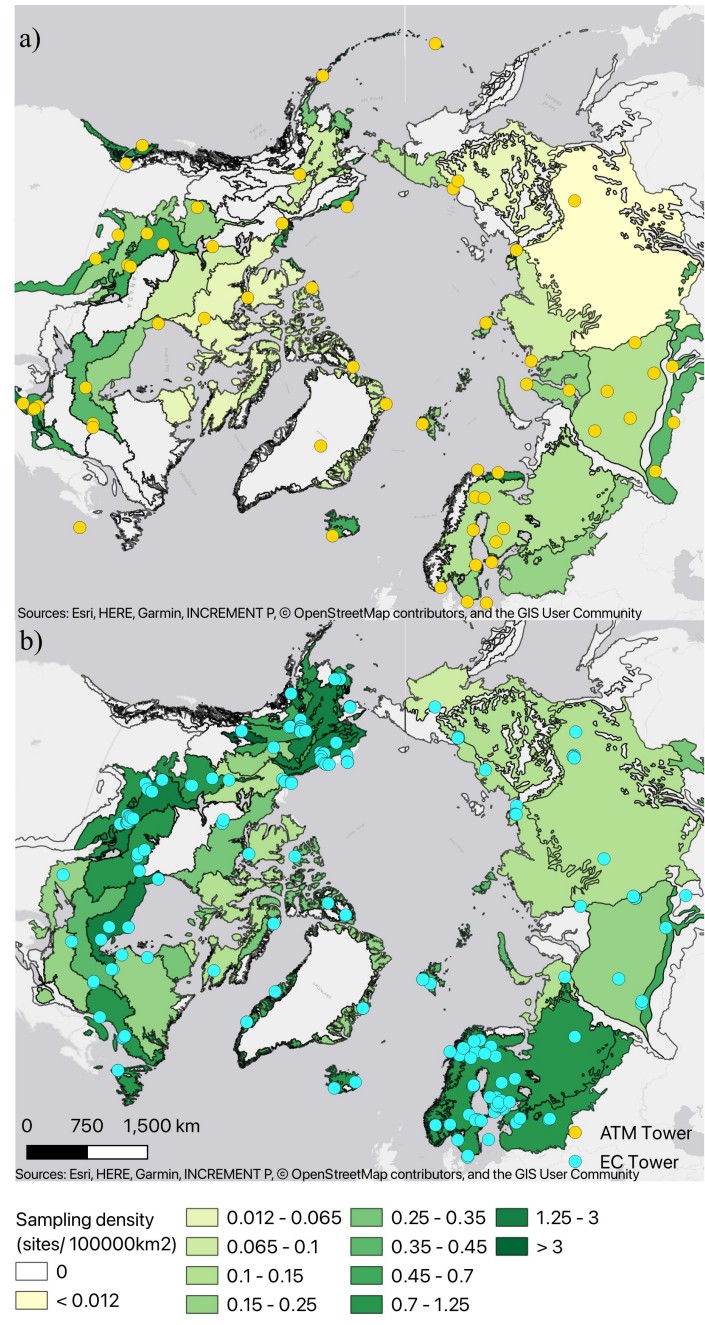

**Figure A1.** Density of study sites for atmospheric towers (a) and eddy covariance towers (b) within ecoregions across high northern latitudes. Numbers are normalised to ecoregion area. Ecoregions were defined based on Olson et al. (2001).

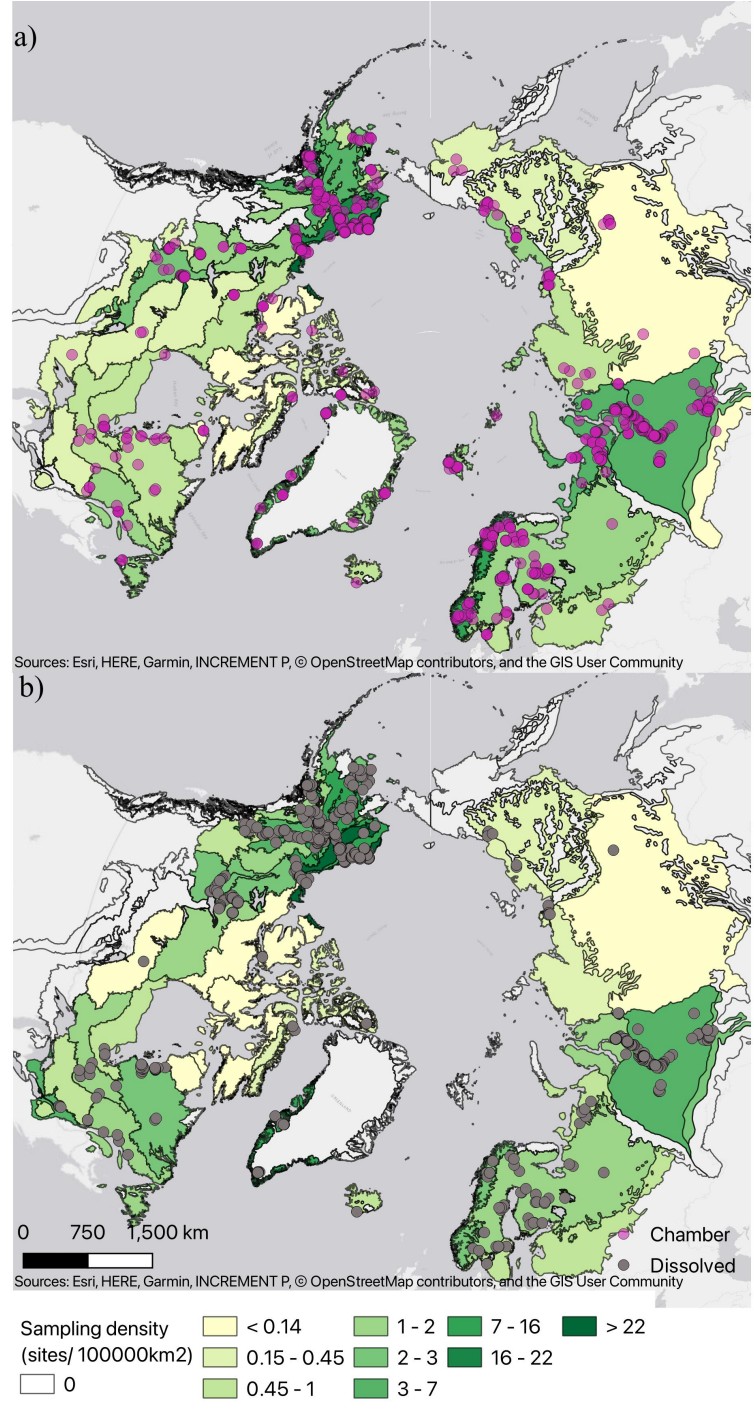

**Figure A2.** Density of study sites for flux chambers (a) and dissolved gases (b) within ecoregions across high northern latitudes. Numbers are normalised to ecoregion area. Ecoregions were defined based on Olson et al. (2001).

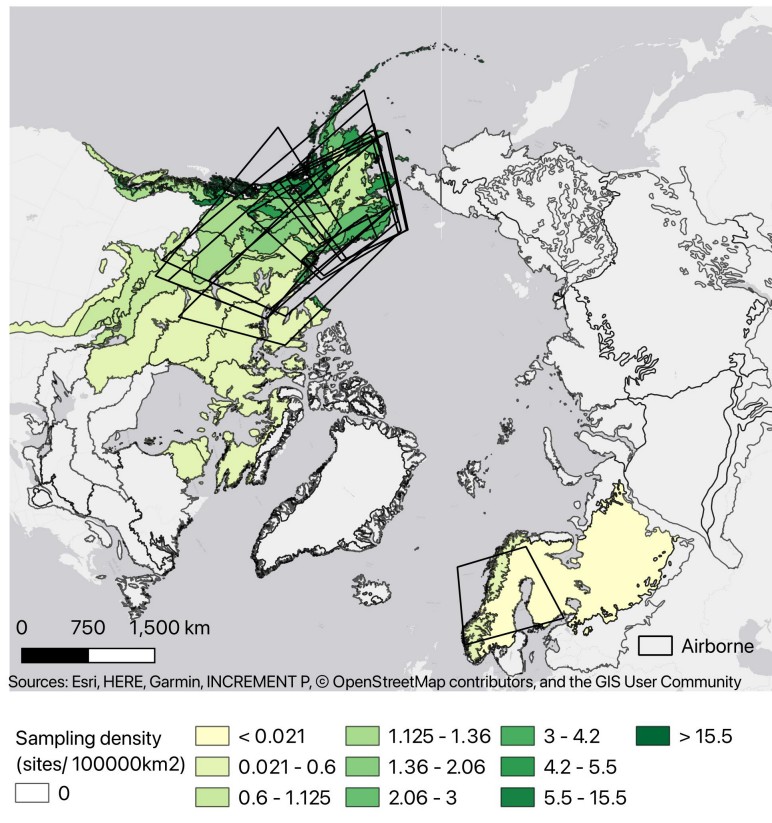

**Figure A3.** Density of study sites for airborne platforms within ecoregions across high northern latitudes. Numbers are normalised to ecoregion area. Ecoregions were defined based on Olson et al. (2001).

*Author contributions.* MMTAP, GC, EAGS, and MG conceived the original version of the online mapping tool. JV, MMTAP, LSB, AB, KI, MK, MM, GC, KA, AMV, and IW contributed to data provision. Data were jointly curated by JV, MMTAP, LSB, AB, KI, MS, MM, GC, and AMV. The formal analysis was conducted by JV, MMTAP, LSB, AB, KI, MS, and AMV, with EAGS, and MG responsible for funding acquisition and supervision. Software was developed and maintained by MMTAP, MS, and GC, while JV, MMTAP, KI, and MS contributed to data visualisation. JV, LSB, AB, KI, and MG wrote the original manuscript. All authors reviewed the manuscript.

*Competing interests.* The authors declare that they have no conflict of interest.

*Acknowledgements.* The authors acknowledge funding from the European Research Council (ERC synergy project Q-Arctic, grant agreement no. 951288), from the European Commission under the Horizon Europe (GreenFeedback project, grant agreement no. 101056921) and Horizon 2020 (INTAROS project, grant agreement no. 727890) programmes, from the German Ministry of Research and Education (MO-

MENT project, support code: 03F0931G), from the Gordon and Betty Moore Foundation (grant no. 8414) and funding catalysed by the TED
Audacious Project (Permafrost Pathways). Funding to EAGS was provided through NSF PLR Arctic System Science Research Networking
Activities (RNA) grant no. 1931333, and the Minderoo Foundation. We acknowledge the usage of deepL Write to improve text passages for
parts of the manuscript.

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
