# Peer review of "ARGO: ARctic greenhouse Gas Observation metadata version 1"

_Earth System Science Data, 2024_

## Author Response (AR1)

**Responses to reviewers**

**February 17, 2025**

**1 Response to reviewer 1**

We would like to thank the reviewer for the kind review, and for sharing user-experience of the associated online tool which will significantly help us improve it. We address the comments below. Comments are initiated by "C:" and our answers by "A:".

C: The authors compiled an extensive resource to locating arctic GHG data. It is impressive in scope and thoroughness. This data compilation will be extremely useful. The online tool is friendly and easy to use. Overall, this is a great metadata set. I have a few comments related to user experience and potentially making the dataset even more user friendly. I know that it is impossible to identify all data existing, and the authors have done a heroic task identifying many datasets. Nonetheless, you skipped some. I understand that no list can ever be all inclusive, but if not too late, consider adding ESS-DiVE. It is a very good data source with hundreds of arctic datasets. Some of them (but not most) are also available through ORNL-DAAC. For example one of tens of ESS-DiVE datasets you do not link: https://data.ess-dive.lbl.gov/datasets/doi:10.5440/1765733

A: Thank you for suggesting another dataset to add. We will add the mentioned dataset and check the related repositories. We will regularly work on updating the dataset in the future.

C: There's a bit of a mismatch between the map and the table in the online tool. Ideally, clicking a site (or dataset marker) on the map should then display or highlight the same site (dataset) in the table, and similarly, clicking a line in the table should highlight a site on the map. However, it appears that the map and table are not connected.

A: This is a great suggestion. We will look into it and try to work on more dynamically linking the map and data table shown within the online tool. We agree that this is currently not perfect and generally plan to work on functionality upgrades to the online tool in the future.

C: A related but different problem – the links to the dataset in the map and the table are not the same. For example – in the table the "link" buttons to the ABOVE_Arctic-CAP datasets (lines 5-10) generate an error (oddly the free

text links in the "data availability" columns, do work), while the map provides a working link.

A: We would like to thank the reviewer for taking the time to thoroughly check this. The link to the repository will be added to the data availability link tab which was causing this issue.

C: Data frustration – the metadata dataset is a fantastic resource. But I assume that the primary usage of it is to locate data. I tested the experience of a user trying to follow the dataset to locate and obtain data. In many cases it works great, but some cases are frustrating. I categorized these frustrations to types, and bring examples: Links to large datasets take a large effort to dig out the specific data that is listed in a particular entry in this metadata. I do not know that I have the solution for that. However, in some cases, they do not lead to any actual data (or I wasn't smart enough to find it). For example: ATQ-207 https://doi.org/10.1038/s41558-017-0066-947 Abbotsford – I followed the link but only found a "site report" and could not figure out how to get the actual data.

A: Thanks for bringing this up. This is a general issue in the research community and not all data will be publicly available by the original authors. That is why we think this meta-dataset is useful in compiling a comprehensive overview of measurements conducted so far. Anyone who is interested to obtain data that is unpublished can at least reach out to the respective contact person and ask for the data themselves. In the particular case that was mentioned (ATQ-207), the reference of the publication is given so that the ARGO user can find further information there. We will implement some clarification in the mapping tool table, so that next to the reference, a link to additional information that goes beyond the publication can be found, as well as the data link which should lead to a place where data can be downloaded from. With those resources together, we try to facilitate access to further information and data. However, it is possible that in some cases none of this information is given. In the case of site ATQ-207, some data is available in the supplement of the paper. Since there is no direct link leading to the supplement, the user will have to find the data themselves through opening the link to the publication. Unfortunately, we cannot provide detailed information for each publication how data can be accessed, and by providing links to the publication, possibly additional information and data portals, we think that we reach the maximum of our capabilities. For more recent and future publications, this issue will likely diminish since many journals require publication of datasets alongside the manuscript.

C: Many entries link to a paper. In such cases there is no "link". I suggest adding the link to the papers DOI or the supplementary dataset that include the actual data as a link button, at the "link" column. In the "data availability" column, you can add the table number within the linked document where the data can be found, to facilitate finding the data more easily.

A: We will change the "Reference" column to give "Reference_Short" in the online tool. This includes the DOI/link to the respective publication for those

measurements where a publication is available. As mentioned in the previous answer, we will further add a column with links to additional information in the table of the online tool. Unfortunately, it is not possible to add instructions for each individual publication on how to access the data or point to the location of data within a manuscript or supplementary file. However, we will add some general instructions in the manuscript to help the user find information more easily.

C: Some papers do not make the data available. For example, lines 58-59 Abisko (which by-the-way, as far as I could see, appear to be the same and I'm not sure why they are listed as two separate entries). I followed the url to the paper and managed to download it through my university library (another cause of user frustration – datasets not free, at least not to everyone), BUT, there was no data anywhere (tables only show long-term averages, no supplementary material). In cases such as these, if you have personal communications with authors for data that were not made available yet, I suggest publishing the data here and a link to a zenodo dataset. Otherwise, consider not listing data sources that do not actually provide data.

A: First of all, thank you for spotting the duplicate (Abisko). We will remove it. As mentioned before, we would still like to add all observations made in the study area, even if unpublished, to realistically broadcast areas that have been studied. Where data is not published, the user of the tool has the opportunity to contact the respective scientists to ask details about the observations undertaken at a specific site. We cannot publish data that was collected by others. Nonetheless, we want to make hidden data visible, or at least point out where data have been collected and would like to refrain from removing data entries without published datasets for this reason. As mentioned before, we give further information in a clearer way in the updated online tool including the doi to the reference, a link with additional information and a link to data repositories.

C: In the maps, you mark some datasets as a point and others as a polygon. As far as I can see, this is done only for airborne datasets. That is a good way to address datasets that offer multiple sites in a particular region. For example, the ABOVE_Arctic-Cap datasets. In the table, these datasets are listed as a particular exact long-lat. Not sure how that particular location was selected, but I suggest enabling a list of long-lats, a link to a polygon or table of points, or a rectangular range (min-max long-lat) in the longitude and latitude columns for datasets that provide data for multiple locations (sub-sites).

A: For the airborne datasets, we list the centroid of the polygon in the table of the online tool. Because of limited space in the online tool table, we chose to only display necessary columns. However, the ARGO meta-dataset (the version that is published here and the one that can be downloaded from the online tool) includes the northern, eastern, western and southern bounds of the polygons for airborne measurements.

C: In other datasets that provide multiple sites, you break each entry within these datasets to a different entry in your metadata table. For example, entries 22-28 for ATQ (this paper is listed many more times, as ATQ is just one of several sites it provides data for multiple locations in the same tables within the paper). I think it'll make sense to combine these, similarly to the airborne datasets, such that, at least data of the same type (e.g., chambers) that come from a single source table, are listed once as a single entry for a spatial range.

A: In some cases, it makes sense to merge data, e.g. for airborne data, where the same group with the same equipment made measurements within a single campaign over a large area. However, for measurements that are rather focused on point measurements, and information about locations (latitude and longitude) are available, or measurements were conducted by different groups or at different times and may thus be related to different publications, we think it is useful to list the individual measurements. We are aware that this approach may bias site counts among measurement platforms, but we also think that we better represent the nature of the types of measurements this way. If a user of the ARGO dataset currently wishes to cluster the data, the data can be downloaded and for example be grouped by reference, years, site name, etc. We will think about how to integrate a direct clustering in the tool for a future upgrade.

**2 Response to reviewer 2**

We would like to thank the reviewer for the constructive feedback to our manuscript. By addressing the suggested comments, the manuscript and associated online tool will significantly improve. Below, we copied the original comments (initiated by "C:") and added our responses marked by "A:".

C: The authors present a database and associated online tool gathering and standardizing metadata for many Arctic (and arctic-adjacent) greenhouse gas measurements. The online tool allows for filtering on basic metadata and yields .csv-formatted spreadsheets of metadata for each platform type. Basic metadata are provided for all sites, including data links and PI contacts. More detailed platform-specific metadata are also included. In the manuscript, a short description of the overall framework is presented. Longer descriptions of each platform are included, as well as brief analyses addressing temporal, methodological, and spatial patterns in the measurements. Some gaps in measurement are identified. Overall I believe this is a useful tool, if properly maintained. It is generally well-described. The overview of ghg monitoring approaches was particularly well-written and helpful; this would be a useful paper to send to new students to acquaint themselves with ghg measurements and initiate a starter project. The writing quality was inconsistent, however, with some very good sections and some poor sections; the bulleted sections need the most revision. I suggest a final writing edit by a single experienced writer to resolve the multiple voices present.

A: Thanks for bringing this up. We will review the manuscript accordingly, making sure that language and writing style will be consistent.

C: In my opinion, the weakest part of this manuscript is the absence of a clear plan for database maintenance. Without dedicated effort, this database will lose its usefulness fairly quickly. The authors should propose a specific set of tasks for maintenance and a (loose) schedule for accomplishing them. Ideally, recommend responsible personnel, or at least decide on it internally to ensure someone takes responsibility.

A: This is a good point. The online tool that was initiated in 2018 and was since then expanded and upgraded will continue to be actively maintained as stated in the manuscript, meaning that responsible persons within the ARGO team will collect and upload new data at regular (3-6 month) intervals. The "frozen-in-time" meta-dataset that we publish and describe here represents the metadata now, while the most up-to-date version of this metadata can be found in the online tool in the future. We will add this information to the conclusions section in the revised manuscript.

C: Another weakness is the brief and somewhat irregular discussion of data gaps. This is not a metanalysis, but it seems like a missed opportunity to have this very nicely categorized dataset and make so few points about opportunities for improvement. The identification of spatial gaps is useful, but more information about temporal gaps (e.g. most campaigns ¡1year?) or ecological gaps (e.g. wet tundra more highly studied than alpine tundra?) would fill out the picture. Systematically categorizing these by type of gap – spatiotemporal, platform, or gas species – would aid in organization.

A: Thanks for this suggestion. We will add further discussion of data gaps, and extend Section 6 of the revised version of this manuscript.

C: Line items below: 28: Tough sentence to parse. My suggestion with minimal change: "However, the vast size of the Arctic region, in combination with logistical challenges of harsh climate conditions and scarce infrastructure, has permitted the establishment of only sparse observational networks."

A: We will change the sentence as suggested by the reviewer.

C: 34 – 37: It seems strange to put the example networks and the papers describing them together. Consider mentioning the networks in the flow of the sentence and then placing all journal articles in a citation at the end.

A: We will change the lists of networks and the associated citations as suggested by the reviewer.

C: 82: flexible → flexibly;

A: This will be changed.

C: 82 - 83: "allowing to include... biome types" reads strangely. Possible revision: "Allowing for the inclusion of more southerly sites to reduce data gaps for certain biomes and regions". Consider adding a sentence here referring to a later discussion of what biomes and regions those will be.

A: We will change the sentence as suggested by the reviewer. We will also add a reference to the discussion already presented in section 3 regarding the inclusion of southerly sites.

C: 84 – 86 "We distinguish... and researchers)" also reads strangely. Possible revision: "We categorize the ecosystem types represented by a study site as barren, cropland, forest, grassland, lake, ocean, reservoir, river, shrubland, tundra, urban, and wetland based on the associated publications or input from site operators and researchers."

A: We will change the sentence as suggested by the reviewer.

C: 90 – 91 "To categorize study periods... ice-covered season (November–April)" also reads strangely. Please revise

A: We will simplify the sentence to 'To categorise study periods, we distinguish between the growing season (labeled 'summer', months May–October) and the snow- and ice-covered season (labeled 'winter', months November–April).'

C: 103: ocean-based → marine

A: This will be adapted.

C: 133: Add more citations to cover time period between 1990s and today, e.g. Foken 2011 (book), Baldocchi 2020.

A: We will add more up-to-date references for the eddy-covariance method.

C: 194: and allow to assess → allow for the assessment of

A: We will adapt this.

C: 202: Besides,... → In addition,...

A: We will follow the reviewer's suggestion and revise the text accordingly.

C: Figure 3: Explicitly mention that the "sawtooth" appearance of the data is due to seasonal differences in availability. Alternatively, have two traces for each color/data type: one representing all-year data (perhaps solid) and one representing growing season data (perhaps dashed).

A: We will address this to improve the appearance of the figure.

C: 266: PI submission =/= accurate! Reword to state almost the opposite: submitted data/metadata are taken as-is from PIs and accuracy is not guaranteed.

A: We agree that a direct contact with a PI does not guarantee that the provided data is perfect. Still, we are talking about metadata here, e.g. instruments used, observation periods defined, etc. Our experience in interacting with

site PIs is that those persons who actually took the time to put together information specifically for our purposes, and were willing to discuss them, actually took great care that the provided information is accurate, and complete. We will therefore slightly change the wording here, but not change the overall message: 'These surveys provide a direct link between site operators or researchers and the meta-data provided in ARGO, and both the detailed feedback provided on different aspects of site operation as well as the option to discuss unclear information directly contributed to an improved accuracy of the relevant data in the ARGO database.'

C: 268-269: Unclear sentence. Is the idea that PIs will check on their meta-data regularly? I think this is unlikely unless there is a formal updating process driven by the ARGO team
A: What we were trying to say here is that regular use of the tool will increase chances that users with different background expertise will find inconsistencies in the data, and report it to site operators. We will modify the text and clarify: 'As an example, database users have reported inconsistencies in the visualized information to database operators in the past, which improved the accuracy of the provided information.'

C: Figure 5: There is an inconsistency in the way atmospheric towers and airborne measurements are represented here. Both are previously stated not to have an assigned ecosystem type, but to be assumed terrestrial for "further analysis" (line 103 – 105). It looks like atmospheric towers are assumed to be terrestrial in this figure (all counts in the "terrestrial" category) while airborne platforms are assumed to be neither (all zeroes across both categories). Please resolve or explain.
A: Thanks for spotting that. Both, atmospheric towers and airborne measurements should be assigned the terrestrial class for simplicity, because both cover a large range and are not usually aimed to distinguish fluxes between ecosystems. The figure will be revised.

C: 290: This is a strange comparison because "airborne" and "N2O" are a platform and a GHG species respectively. Additionally, those observations are sparse for different reasons. Open this point differently. Consider also that airborne observations are very information-rich; one chamber measurement offers orders of magnitude less information than one airborne campaign.
A: These are really good points. We will completely restructure Section 6 to organise the discussion of data gaps in a more logical way. Regarding the description of airborne campaigns, the text will be changed to 'Airborne observations: Even though for this platform only few datasets are available, they consist of extended flight legs each, covering large areas with very detailed and information-rich observations. From the pan-Arctic perspective, the main gap consists in the uneven spatial distribution: From the 14 airborne datasets currently listed in ARGO, just one is not focusing on Alaska and Northwestern Canada. Moreover, no campaigns were conducted during wintertime.'

C: 297-299: This is interesting and important but narrow. Consider expanding this paragraph to add more context, supported by studies, for these information gaps. For example, resource limitations in certain countries, accessibility concerns, or technological gaps.

A: The cited short paragraph was specifically aiming at the disruption to Arctic research imposed by the lack of communication between the Russian and Western science communities as a consequence of the war in Ukraine. We will add more context here, adding a sentence that cites one of the listed studies: 'For example, for the pan-Arctic eddy-covariance network Schuur et al. (2024) quantified a loss of spatial representativeness from 0.55 to 0.36 (minus 35 percent) linked to the missing access to 27 stations situated in Russia. Even a targeted investment into new sites in e.g. North America could only make up for about one third of this information loss, emphasizing the need to, when the time is right, develop strategies to keep Russian sites operational and facilitate data exchange and communication for the benefit of Arctic climate research.' Regional data gaps have been discussed already at other sections of the manuscript, and will not be included once more at this point.

C: Comments on the online tool (for authors' information; not necessary to address for MS recommendation): - I appreciate the CSV output (rather than .nc or .xlsx); this will improve accessibility for less code-savvy users (and those of us who keep accidentally mangling timestamps with excel) - In the online site list, the reference column explodes the row height when many authors are included. Can this be fixed? The Reference_short column seems like it could substitute. - Consider adding reference and/or data DOI column(s) where applicable. - The output is helpfully data-rich but I doubt the accuracy of some information. For instance, almost all flux towers will have a radiometer and Temp/RH sensor; this column is NA for many towers. I suppose this is the best you can do with limited info, but I could see users mistakenly throwing away usable data by subsetting on such columns. - It would be neat to have more sophisticated spatial subsetting, for instance, retrieving measurements within a user-uploaded .kml or .shp.

A: Thank you for adding further comments on the online tool. We will add the "Reference_Short" column instead of the long citation. This also includes the doi. Regarding the remaining comments, we will try to implement them in future upgrades.